# Contrasting Differences in Responses of Streamflow Regimes between Reforestation and Fruit Tree Planting in a Subtropical Watershed of China

**Zhipeng Xu [1], Wenfei Liu [1],\*, Xiaohua Wei [2], Houbao Fan [1], Yizao Ge [1], Guanpeng Chen [1] and Jin Xu [1]**

[1] Jiangxi Province Key Laboratory for Restoration of Degraded Ecosystems & Watershed Ecohydrology, Nanchang Institute of Technology, Nanchang 330099, China; xuzhipeng34@163.com (Z.X.); hbfan@nit.edu.cn (H.F.); geyizao@163.com (Y.G.); 15279196557@163.com (G.C.); 18296150346@163.com (J.X.)

[2] Earth and Environmental Science Department, University of British Columbia (Okanagan), 1177 Research Road, Kelowna, BC V1V 1V7, Canada; adam.wei@ubc.ca

\* Correspondence: 2007992987@nit.edu.cn; Tel.: +86-0791-82085311

**Abstract:** Fruit tree planting is a common practice for alleviating poverty and restoring degraded environment in developing countries. Yet, its environmental effects are rarely assessed. The Jiujushui watershed (261.4 km$^2$), located in the subtropical Jiangxi Province of China, was selected to assess responses of several flow regime components on both reforestation and fruit tree planting. Three periods of forest changes, including a reference (1961 to 1985), reforestation (1986 to 2000) and fruit tree planting (2001 to 2016) were identified for assessment. Results suggest that the reforestation significantly decreased the average magnitude of high flow by 8.78%, and shortened high flow duration by 2.2 days compared with the reference. In contrast, fruit tree planting significantly increased the average magnitude of high flow by 27.43%. For low flows, reforestation significantly increased the average magnitude by 46.38%, and shortened low flow duration by 8.8 days, while the fruit tree planting had no significant impact on any flow regime components of low flows. We conclude that reforestation had positive impacts on high and low flows, while to our surprise, fruit tree planting had negative effects on high flows, suggesting that large areas of fruit tree planting may potentially become an important driver for some negative hydrological effects in our study area.

**Keywords:** reforestation; fruit tree planting; flow regimes; high flows; low flows

## 1. Introduction

The relationship between forest cover changes and streamflow has long been a heated topic in forested regions [1–3]. Over the past decades, numerous reviews have been made on the effects of forest change on annual mean flow [4–9]. However, research on the impacts of forest change on flow regimes is rather limited [10–13]. Flow regime is composed of five elements: magnitude, duration, timing, variability and frequency [11,14], and the alteration of any element can affect aquatic habitat and biodiversity, as well as ecosystem integrity [15–17]. For example, changes in magnitude and frequency are likely to affect the transport of organic matter and sediments, while changes in flow timing and duration could lead to interference of salmon spawning, and consequently salmon life cycle [11,18]. Therefore, there is a critical need to study how forest cover changes may affect flow regime components where large forest cover change occurs.

Reforestation or afforestation is considered as one of the effective measures to address environmental degradation and climate change impact [19]. To alleviate poverty and environmental degradation, many rural communities in China, particularly in southern China, often grow fruit trees to increase short-term economic benefits and prevent soil erosion. However, forest structures and management strategies resulting from fruit trees and nature forest stands are different. It is still not clear whether fruit tree stands have as similar hydrological functions as natural forests do. Due to more frequent floods and drought events occurring in reforested regions [20], there is a growing concern over the possible negative effects of large-scale fruit tree planting on hydrological functioning. As such, understanding this research topic can support watershed management decisions regarding the relationships between reforestation and water resources.

High and low flows play an important role in the structural composition and function maintenance of riverine aquatic ecosystems by shaping the geomorphologic features of channels and floodplains [10,11,17]. High flow is an indicator of the intensity of floods, and of great significance for public safety. Similarly, low flow regimes are closely related to the functions and structures of riparian plant species [21,22]. Assessing the effectiveness of different forest restoration strategies such as reforestation and fruit trees planting on high and low flows can provide important insights into understanding hydrological processes in forested watersheds.

Jiujushui watershed (261.4 km$^2$) is located in the subtropical region of China. Over the past decades, it has experienced dramatic changes in forest cover, including a forest degradation period in 1960s, reforestation from 1986 to 2000, and fruit tree planting since 2001. In particular, the area of fruit tree planting has been greatly increased over the past 10 years. Such a dramatic forest cover change in the watershed provides a unique opportunity for studying the effects of various reforestation strategies on hydrology. Therefore, the main objective of this study was to the examine whether reforestation and fruit tree planting have led to significant changes in high and low flow regimes in the Jiujushui watershed, and if so, how big the changes have been.

## 2. Materials and Methods

### 2.1. Study Area

The Jiujushui watershed, located in the upper reach of Fu River, is one of the main tributaries of the Poyang Lake basin of Jiangxi Province in the southeast of China (Figure 1). The watershed has a drainage area of 261.4 km$^2$ with the range of slope from 0° to 50°, a main channel drainage length of 41.8 km and an average elevation of 231 m (Figure A1a,b). Red soil, yellow-red soil and mountain yellow soil are the main soil types. Red soil is normally distributed in hilly areas with elevations as low as 500 m below sea level, while yellow-red soil is distributed in areas with an altitude of 500 to 800 m and mountain yellow soil is located in areas with an altitude of more than 800 m. Furthermore, red soil varies with soil depth: A horizon, B horizon and C horizon (Csv). Within the humid subtropical monsoon zone, Jiujushui watershed received an average annual precipitation of 1780 mm between 1961 and 2016, with 855 mm (48.0%) in the wet season from April to June and 225 mm (12.6%) in the dry season from September to November. Annual mean, maximum and minimum temperatures are 18.4 °C, 34.8 °C (in July) and 2.8 °C (in January), respectively (Figure 2). The major land cover types include forest land, farmland and urban. Based on historical land use data, the changes in farmland and urban only accounted for <3.5% (1962–2006) and 0.2% (1996–2005) of the watershed area, respectively, while forest cover change occurred from 36.4% to 77.1% between 1985 and 2016 (reforestation and fruit tree planting).

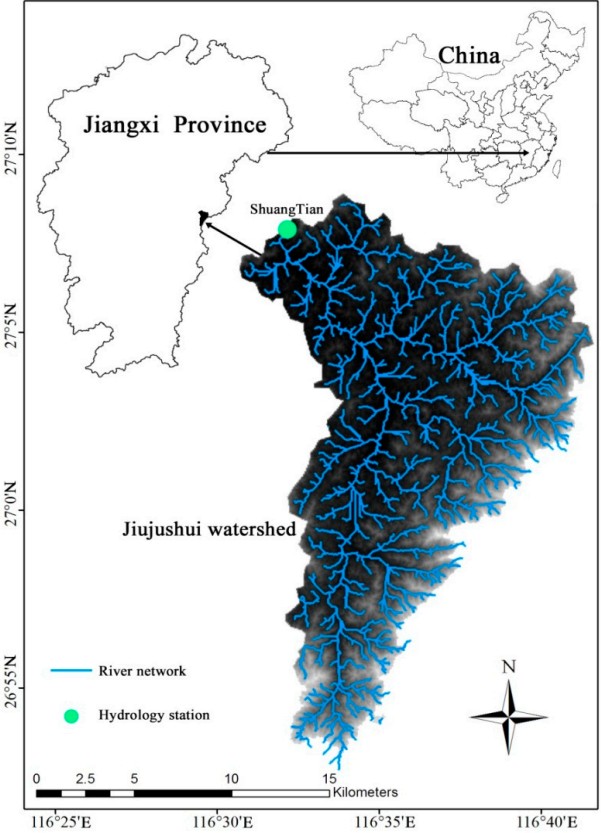

**Figure 1.** Location of the Jiujushui watershed.

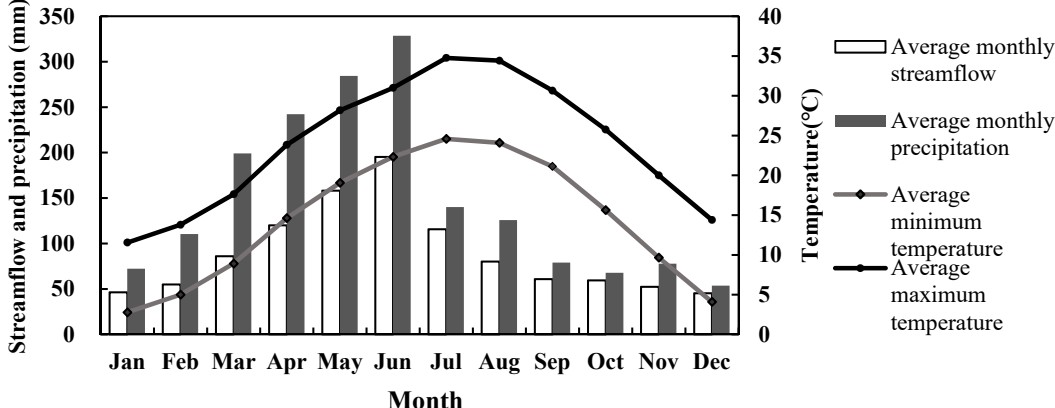

**Figure 2.** Average monthly precipitation and streamflow from 1961 to 2016, with maximum and minimum temperatures from 1961 to 2011 in the Jiujushui watershed.

### 2.2. Data Collection

Stream flow data (1961–2016) used in this study were collected from Shuangtian hydrometric station (Station ID: 62406200) as part of the Chinese National Hydrological Network. Maximum and minimum daily flows were 396 m$^3$s$^{-1}$ (2002) and 0.16 m$^3$s$^{-1}$ (1963), respectively. The annual streamflow varied from 414.2 mm in 1963 to 1820.5 mm in 2016, with an annual mean discharge of 1080.2 mm from 1961 to 2016 (Figure A2). Historical climate data from 1961 to 2011, including daily precipitation and daily maximum, mean and minimum temperatures, were obtained from the Climate Center of Jiangxi Province.

Forest data (forest coverage data and area of fruit tree planting) were obtained from historical forest resources inventory in Nanfeng County. The major forest types included protection forest,

timber forest and economic forest, among which *Pinus massoniana*, *Cunninghamia lanceolata*, citrus and *Phyllostachys heterocycla* (Carr.) *Mitford cv. pubescens* were dominant species in plantation forests. The typical planting density for fruit trees (citrus) in the study watershed is 3.5 m × 4.5 m.

### 2.3. Defining the Periods of Forest Changes

The watershed experienced two distinct forest cover changes over the period of 1961 to 2016. From 1961 to 1985, forest coverage in Jiujushui watershed declined by only 6.3%. Thereafter, a sharp increase of 40.7% from 1986 to 2016 attributed to the Sloping Land Conversion and Mountain-River-Lake Ecological programs in Jiangxi Province, during which fruit tree planting exponentially expanded after 2000 (Figure 3). Therefore, the whole research period was divided into three sub-periods—forest degradation (or the reference period from 1961 to 1985), reforestation (1986–2000) and fruit tree planting (2001–2016)—based on the historical forest changes. It should be noted that the forest degradation period includes 6.3% of forest cover change. Stednick [2] stated that at least of 10~20% change in the watershed area is needed to produce significant hydrological changes. In addition, such a minor change occurred over the 20 years, without significant hydrological effects, that the forest degradation period (1961 to 1985) was treated as the reference or baseline period.

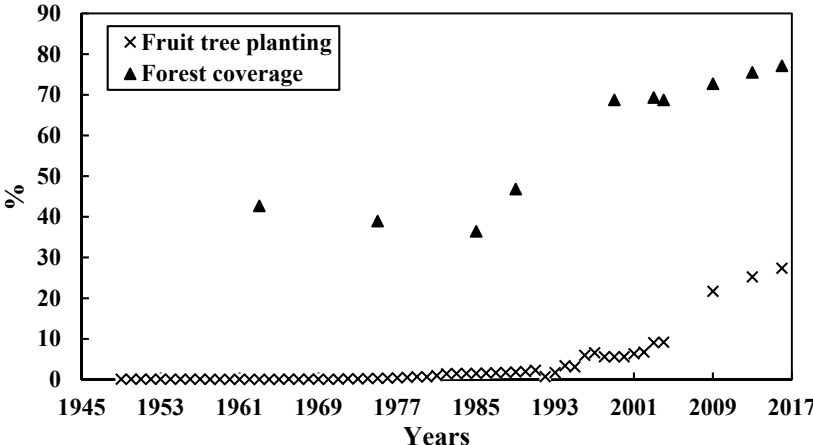

**Figure 3.** Forest changes (forest coverage and fruit tree planting) in the Jiujushui watershed.

### 2.4. Defining High and Low Flow Regimes

In this study, high and low flows are defined as the flows that are equal or more than $Q_{5\%}$, or equal or less than $Q_{95\%}$, in a given year, respectively, through flow duration curves, which represent the relationship between discharges and percentages of discharges below or exceed certain levels for a given time period [10,23,24].

The flow regime includes five elements, which are magnitude, timing, duration, variability and frequency [11]. In this study, magnitude refers to the total amount of water moving through the outlet of the watershed in a given time. The time interval between rainfall peaking and flow peaking represents the timing of high flow, while timing for low flow refers to the average date of low flow occurrence in a year using the paired-wise approach (see the next section) [20]. Duration is defined as the number of days in which daily flow exceeded or was below a given magnitude: high flows refers to the number of days when daily flow exceeded or equaled median value in a given year, while duration of low flows refers to the number of days with daily flow below or equal to median value in a given year. Variability is denoted by the coefficient of variation (CV). Frequency represents how often flow exceeded or was below a given magnitude or return period of high and low flows. Using flood frequency analysis combined with Log-Pearson Type III for analyzing return periods [20,25,26], we divided the return periods of high and low flows into four types: $T_r \leq 1$, $1 < T_r \leq 2$, $2 < T_r \leq 5$ and $5 < T_r \leq 10$ according to the data, where $T_r$ represents the return period.

*2.5. Elimination the Effect of Climate Factors on High and Low Flows with The Paired-Year Approach*

To minimize the effects of climate variability on high and low flows, the paired-wise approach was used [10,23,24,27], which compares the flow regimes in forest change periods against the reference period under similar climate conditions. As such, the differences between two periods are mainly attributed to forest cover change. In this study, seasonal (wet = April–June, and dry = December–February) and annual precipitation, mean, maximum and minimum temperatures and wind speed were selected as proxies to represent climate conditions over these periods. Firstly, Kendall's Tau and Spearman's Rank were conducted to examine statistical relationships between seasonal and annual climate variables, as well as hydrological variables. As shown in Table A1, annual precipitation (P) and wet-season precipitation ($P_w$), as well as maximum ($T_{maxw}$) and average temperature ($T_{avew}$) in the wet season were significantly correlated with high flows, while P, $P_w$, $T_{maxw}$ and $T_{max}$ were significantly correlated with low flows, where subscript w denotes wet season. Secondly, canonical correlation analysis was used to examine the correlations between two sets of variables, and was elected to determine the highest correlations between sets of climate variables and sets of high and low flows [24]. As a result, $P_w$, $T_{max}$, $T_{maxw}$ and $T_{avew}$ were eventually determined as proxies for similar climate conditions between the reference and reforestation periods during low and high flows (Table A2). Finally, climate variables between the reference and reforestation periods, and between the reference and fruit tree planation periods, were selected (Table A3). It should be noted that high flows are normally associated with storm events. Therefore, the different timings of high flows were selected based on the similarity of storm events in the forest cover change periods (Table A4).

## 3. Results

### 3.1. Responses of High Flows to Reforestation and Fruit Tree Planting

#### 3.1.1. Magnitude

The average magnitude of high flows in the reference period (1961–1985) and reforestation period (1986–2000) was 44.76 $m^3s^{-1}$ and 40.83 $m^3s^{-1}$, respectively. High flows were significantly reduced by 8.78% ($p$ = 0.018) when compared to the reference period (Figure 4a). Conversely, the average magnitude of high flows in the fruit tree planting period (2001–2016) was increased by 27.43% ($p$ = 0.044) in comparison with the reference period (Figure 4b).

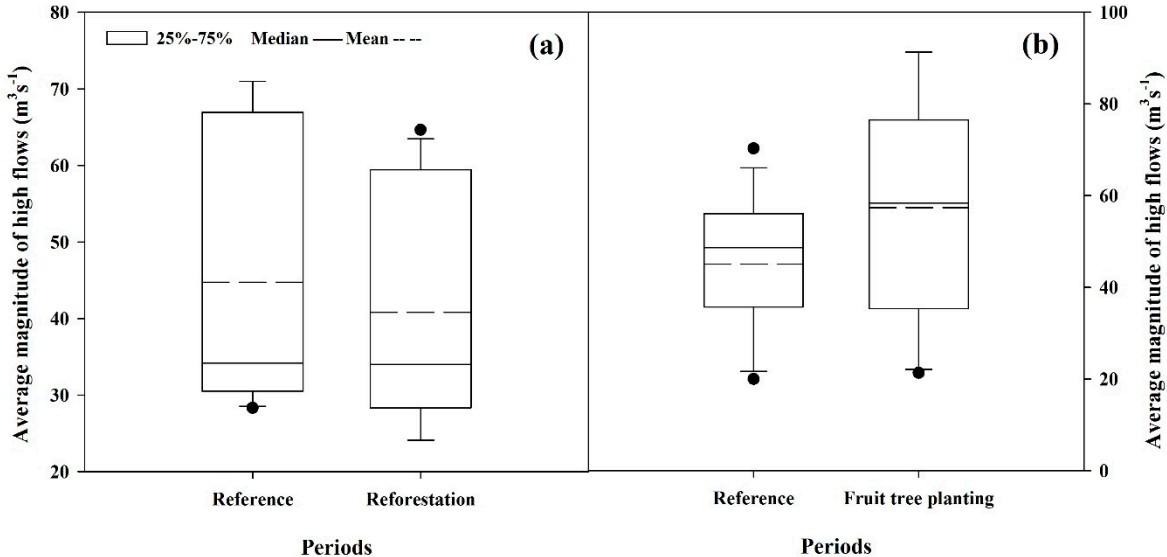

**Figure 4.** Comparison of the magnitudes of high flows (**a**) between the reference and reforestation periods, and (**b**) between the reference and the fruit tree planting periods.

### 3.1.2. Timing

Rainfall events from 25 to 75 mm were selected to find the time intervals between rainfall peaking and flow peaking (Table A4). No statistically significant relationships were detected between forest changes and average timing of high flows in the reforestation period ($p = 0.136$) and the fruit tree planting periods (no delay), respectively.

### 3.1.3. Duration

The analysis from all paired years revealed that the average duration of high flows in the reforestation period was significantly shortened by 2.2 days ($p = 0.033$) than that in the reference period (Figure 5). In contrast, the average duration of high flows was not statistically altered in the fruit tree planting period ($p = 0.235$), but varied with events.

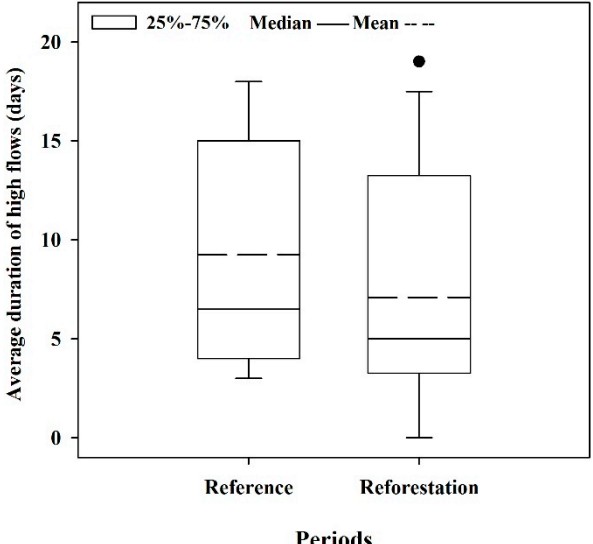

**Figure 5.** Reference period vs. reforestation period: comparison of the duration of high flows.

### 3.1.4. Frequency

Compared with the reference period, the reforestation and fruit tree planting periods had no significant effects on the return periods of $T_r \leq 1$, $1 < T_r \leq 2$, $2 \leq T_r \leq 5$ and $5 \leq T_r \leq 10$ of high flows, respectively ($p = 0.260$ and $p = 0.155$, respectively).

### 3.1.5. Variability

The reforestation and fruit tree planting periods had no statistically significant impacts on the average CV of high flows ($p = 0.911$ and $p = 0.326$, respectively).

### 3.2. Responses of Low Flows to Reforestation and Fruit Tree Planting

### 3.2.1. Magnitude

The average magnitude of low flows in the reforestation period was 46.38% ($p = 0.026$) higher than that in the reference period (Figure 6). In contrast, the magnitude of low flows was not significantly altered in the fruit tree planting period ($p = 0.234$).

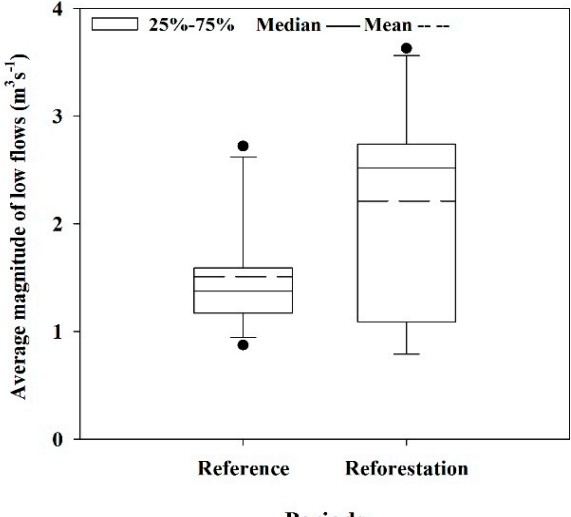

**Figure 6.** Reference period vs. reforestation period: comparison of the average magnitude of low flows.

### 3.2.2. Timing

Reforestation and fruit tree planting had no statistically significant impact on the average timing of low flows ($p = 0.975$ and $p = 0.108$, respectively).

### 3.2.3. Duration

The average duration of low flows in the reforestation period was significantly longer ($p = 0.007$) than that in the reference period. On the contrary, the average duration of low flows was insignificantly related to fruit tree planting ($p = 0.085$). As an example, for the paired years of 1968 and 1994, the daily flows (below or at 1.93 $m^3s^{-1}$) in 1968 (reference year) and in 1994 (reforestation year) were 18 and 0 days, respectively (Figure 7).

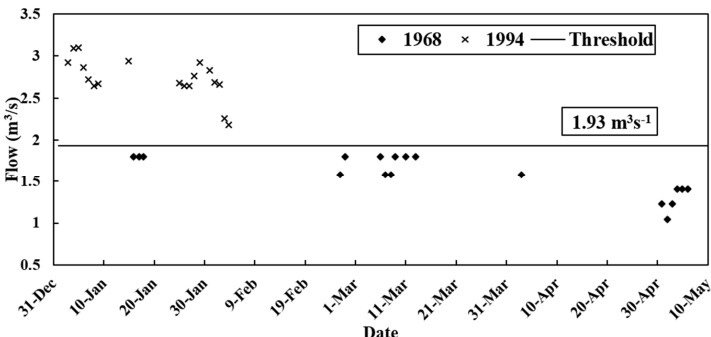

**Figure 7.** Comparison of the duration of low flows between a reference year (1968) and reforestation year (1994) (please refer to the definition of flow duration in the text for further clarification).

### 3.2.4. Frequency

Reforestation and fruit tree planting had insignificant impacts on the return periods ($1 < T_r \leq 2$, $2 < T_r \leq 5$, $5 < T_r \leq 10$) of low flows ($p = 0.231$ and $p = 0.111$, respectively).

### 3.2.5. Variability

The CV of low flows was not significantly related to either reforestation or fruit tree planting, indicating that those two forest practices had similar impacts on the CV of low flows ($p = 0.499$ and $p = 0.689$, respectively).

## 4. Discussion

### 4.1. Effects of Reforestation on High and Low Flows

In the study watershed, reforestation significantly decreased the magnitude and duration of high flows, which is consistent with the general conclusion from other studies [23,28–30]. The reduction of high flows means lower probability of flood occurrence. Reforestation increases leaf area, evapotranspiration and canopy interception of rainfall, and consequently results in reduction of high flows and surface runoff [31]. On the other hand, reforestation enriches the vertical spatial structure of forest ecosystems, abundant understory vegetation and litter layers that can effectively absorb and store water, reducing peak flows [32,33].

Our results also indicated that reforestation had significantly reduced the magnitude and duration of low flows, which are similar to the findings reported in other studies [34–36]. Reforestation can restore the infiltration capacity and water retention ability of soil [23,37,38], increase soil moisture content and groundwater recharge and consequently enhance low flows [39,40]. However, the significant changes in the other flow regime components of low flows were not detected in the reforestation period. This may have been due to the slow forest recovery of soils after severe soil erosion occurred in the reference or forest degradation periods [20].

### 4.2. Effects of Fruit Tree Planting on High and Low Flows

To our surprise, our results indicated that fruit tree planting significantly increased the magnitude of high flows in comparison with the baseline, suggesting that fruit tree planting had negative effects on high flows as the increased high flows produce a higher chance of flooding. Unlike reforestation, fruit tree planting is a distinct planting activity. Fruit tree planting often disturbs the ground surface and reduces surface roughness through intensive land management measures such as soil preparation and removal of understory vegetation and litter layers. As a result, the water-holding capacity of understory vegetation is impaired, which could lead to increased surface runoff [41–44]. Furthermore, removal of weeds and litter can damage the ground surface structure and consequently reduce rainfall infiltration and increase soil and water loss [45].

Our study showed that fruit tree planting had no significant impact on all flow regime components as compared with the reference period. This demonstrates that fruit tree planting did not significantly improve low flows. This is contrary to our expectation, as fruit tree planting is normally expected to play a positive role in soil and water conservation due to the increasing of forest cover. This finding is likely due to intensive land management and resultant reduction of soil infiltration capacity. For instance, some studies demonstrated that the intensive land management in the process of vegetation planting may degrade soil and alter soil infiltration capacity [46], making it impossible for rainfall to infiltrate, and leading to a decrease of soil moisture and groundwater recharge [47].

### 4.3. Contrasted Differences

Both reforestation and fruit tree planting can greatly improve forest coverage. However, their role in flow regimes of high and low flows are different. Our study showed that reforestation had significant and positive effects on the restoration of hydrological processes, while fruit tree planting significantly increased the magnitude of high flows and chance of flooding. Thus, fruit tree planting mainly showed negative effects on flow regimes (Table 1), which is contrary to what we previously anticipated.

The contrasted difference in the responses of flow regimes between the reforestation and the fruit tree planting periods in our study was mainly due to vegetation types and associated management practices [48]. In the study watershed, forest types for reforestation were mainly arbor, while fruit tree species is similar to shrubs due to their similar heights and leaf areas. Lu et al. [49] demonstrated that the runoff magnitude of shrubs was higher, and runoff generation time of shrubs significantly earlier, than arbor. Furthermore, forest stands through reforestation often have well-established understory vegetation and litter layers, while fruit tree planting allows limited understory vegetation and litterfall

due to control of weeds from intensive management. Our finding is consistent with Huang et al. [45], who suggested that single-type citrus orchards and farming can significantly increase surface runoff compared with natural vegetation restoration. Thus, vegetation types and associated management practices are critical to flow regimes

**Table 1.** The effects of forest changes on flow regimes in the Jiujushui watershed.

| Period | | Flow Regime Components | | | | |
|---|---|---|---|---|---|---|
| | | Magnitude | Timing | Duration | Frequency | Variability |
| Reference period vs. reforestation period | High flows | ↓ * | - | ↓ * | - | - |
| | Low flows | ↑ * | - | ↓ ** | - | - |
| Reference period vs. fruit tree planting period | High flows | ↑ * | - | - | - | - |
| | Low flows | - | - | - | - | - |

\* $p \leq 0.05$, \*\* $p \leq 0.01$.

### 4.4. Uncertainty Analysis

There are several uncertainties in this study. Firstly, the paired-wise method has its own strengths and limitations. The current literature indicates that the paired-wise approach is an effective assessment technique for large watersheds (>100 km$^2$). However, its accuracy is largely dependent on the data used to select suitable and comparable pairs. In this study, the combination of Kendall's Tau, Spearman's Rank and canonical correlation over the different seasons ensured that our selected climatic variable was significantly related to both high and low flows. Although annual and seasonal climatic variable were considered, more climate variables in shorter time intervals could be included for better selection in this approach. Secondly, the effects of forest cover change on hydrology are cumulative, which mean that such effects can be prolonged over a long period of time. In this study, averaged hydrological effects between the reference and reforestation or fruit tree planting periods were assessed, which did not differentiate the dynamic and cumulative nature of hydrological effects over the study period. Finally, our study detected significantly negative effect on high flows. This might be related to several mechanisms including site preparation, control of weeds or other human activities. Further studies are needed to understand their relative effects so that the negative hydrological effects caused by fruit tree planting can be minimized. Despite the negative effects on high flow, its effects on the other studied hydrological variables were insignificant in this study, suggesting that the hydrological effects of fruit tree planting may be more complicated than we previously thought, which need more attention.

### 5. Conclusions

Our results show that reforestation has positive effects on high and low flow regimes, while fruit tree planting has negative effects on high flows in our studied watershed. The negative hydrological effects from fruit tree planting suggest that fruit tree planting may not always provide environmental benefits as previously expected, even though it increases forest coverage. To restore soil and water functions in the degradated areas of subtropical regions, caution must be exercised when selecting vegetation types and management practices. It also highlights future studies are needed to fully assess possible contributing mechnisms to increasing of high flows caused by fruit tree planting.

**Author Contributions:** Data curation, H.F.; Investigation, Z.X., Y.G., G.C. and J.X.; Methodology, X.W.; Supervision, W.L.; Writing—original draft, Z.X.; Writing—review & editing, Z.X., W.L. and X.W.

**Acknowledgments:** This work was supported by the National Natural Science Foundation of China (31660234), Jiangxi Provincial Department of Education (GJJ151141) and Scientific Funding by Jiangxi Province (20161BBH80049).

**Conflicts of Interest:** The authors declare no conflict of interest.

## Appendix A

**Table A1.** Correlation analysis between hydrological variables and climate factors.

| Annual | P | | $T_{max}$ | | $T_{min}$ | | $T_{ave}$ | | Wind Speed | |
|---|---|---|---|---|---|---|---|---|---|---|
| | M-K | S | M-K | S | M-K | S | M-K | S | M-K | S |
| High flow | 0.558 ** | 0.767 ** | −0.173 | −0.267 | 0.074 | 0.12 | −0.081 | −0.119 | 0.048 | 0.07 |
| Low flow | 0.516 ** | 0.704 ** | −0.202 * | −0.282 * | 0.103 | 0.148 | −0.105 | −0.148 | 0.04 | 0.027 |
| **Wet seasons** | **$P_w$** | | **$T_{maxw}$** | | **$T_{minw}$** | | **$T_{avew}$** | | **$W_w$** | |
| | M-K | S | M-K | S | M-K | S | M-K | S | M-K | S |
| High flow | 0.640 ** | 0.831 ** | −0.282 ** | −0.352 ** | −0.040 | −0.059 | −0.193 * | −0.246 | −0.051 | −0.054 |
| Low flow | 0.551 ** | 0.739 ** | −0.230 * | −0.287 * | 0.010 | 0.030 | −0.143 | −0.173 | −0.041 | −0.054 |
| **Dry season** | **$P_d$** | | **$T_{maxd}$** | | **$T_{mind}$** | | **$T_{aved}$** | | **$W_d$** | |
| | M-K | S | M-K | S | M-K | S | M-K | S | M-K | S |
| High flow | 0.009 | 0.017 | 0.052 | 0.081 | −0.021 | −0.030 | −0.016 | −0.008 | 0.011 | 0.017 |
| Low flow | −0.092 | −0.142 | −0.040 | −0.068 | −0.117 | −0.170 | −0.092 | −0.118 | 0.057 | 0.081 |

Note: P, Tmax, Tave and Tmin are annual mean precipitation, maximum, average, minimum, respectively; Pw and Pd are wet season precipitation (April–June) and dry season precipitation (December–February). M-K and S refer to the methods of Mann–Kendall and Spearman correlation. Significance level set at 0.05, * $p \leq 0.05$, ** $p \leq 0.01$.

**Table A2.** Canonical correlation analysis between hydrological variables and climate factors.

| | Canonical R | Hydrological Variable Set (High and Low Flows) |
|---|---|---|
| | Set 1 (P, $T_{max,}$) | 0.774 ** |
| | Set 2 (P, $T_{maxw}$, ) | 0.774 ** |
| | Set 3 (P, $T_{avew}$) | 0.774 ** |
| | Set 4 (P, $T_{max}$, $T_{avew}$) | 0.774 ** |
| | Set 5 (P, $T_{maxw}$, $T_{avew}$) | 0.793 ** |
| | Set 6 ((P, $T_{max}$, $T_{maxw}$) | 0.778 ** |
| | Set 7 (P, $T_{max}$, $T_{maxw}$, $T_{avew}$) | 0.795 ** |
| Climate variables sets | Set 8 ($P_w$, $T_{max}$) | 0.842 ** |
| | Set 9 ($P_w$, $T_{maxw}$) | 0.842 ** |
| | Set 10 ($P_w$, $T_{avew}$) | 0.841 ** |
| | Set 11 ($P_w$, $T_{max}$, $T_{avew}$) | 0.846 ** |
| | Set 12 ($P_w$, $T_{maxd}$, $T_{avew}$) | 0.844 ** |
| | Set 13 ((P, $T_{max}$, $T_{maxw}$) | 0.845 ** |
| | Set 14 ($P_w$, $T_{maxw}$, $T_{maxw}$, $T_{avew}$) | 0.847 ** |

* $p \leq 0.05$, ** $p \leq 0.01$.

**Table A3.** Climate and flow variables of paired years in the Jiujushui watershed.

| | Selected Year | Paired Year | $P_w$/mm | | $Q_w$/mm | | $T_{max}$/°C | | $T_{maxw}$/°C | | $T_{avew}$/°C | |
|---|---|---|---|---|---|---|---|---|---|---|---|---|
| | 1968 | 1994 | 1083.1 | 1072.4 | 469.6 | 650.3 | 24.1 | 23.6 | 27.2 | 28.5 | 21.8 | 23.4 |
| | 1979 | 1990 | 609.1 | 614.9 | 278.1 | 284.7 | 24.2 | 23.5 | 26.8 | 27.3 | 21.9 | 22.1 |
| | 1964 | 1986 | 668.3 | 650.8 | 387.8 | 335.4 | 23.7 | 24.0 | 27.7 | 28.2 | 23.1 | 23.1 |
| | 1979 | 1987 | 609.1 | 635.3 | 278.1 | 250.7 | 24.2 | 24.1 | 26.8 | 27.3 | 21.9 | 22.1 |
| | 1968 | 1998 | 1083.1 | 1105.1 | 469.6 | 611.8 | 24.1 | 24.6 | 27.2 | 29.3 | 21.8 | 23.9 |
| Reference period vs. | 1974 | 2000 | 722.2 | 721.6 | 230.7 | 351.7 | 23.6 | 23.3 | 28.5 | 28.0 | 23.0 | 22.8 |
| reforestation period | 1965 | 1993 | 865.6 | 790.4 | 349.2 | 344.1 | 23.6 | 23.6 | 26.4 | 26.9 | 21.4 | 22.1 |
| | 1966 | 1996 | 874.3 | 852.8 | 512.5 | 416.8 | 24.3 | 23.6 | 27.0 | 26.8 | 21.8 | 21.7 |
| | 1982 | 1994 | 978.3 | 1072.4 | 669.0 | 650.3 | 23.3 | 23.6 | 26.8 | 28.5 | 21.6 | 23.4 |
| | 1981 | 1995 | 857.9 | 849.0 | 653.3 | 431.4 | 23.3 | 23.5 | 27.2 | 27.2 | 22.1 | 22.3 |
| | 1972 | 1993 | 819.9 | 790.4 | 283.8 | 344.1 | 23.3 | 23.6 | 26.9 | 26.9 | 21.9 | 22.1 |
| | 1978 | 1987 | 689.4 | 635.3 | 347.9 | 250.7 | 24.2 | 24.1 | 27.1 | 27.3 | 21.9 | 22.1 |

**Table A3.** *Cont.*

| | Selected Year | Paired Year | $P_w$/mm | | $Q_w$/mm | | $T_{max}$/°C | | $T_{maxw}$/°C | | $T_{avew}$/°C | |
|---|---|---|---|---|---|---|---|---|---|---|---|---|
| | 1975 | 2001 | 1090.4 | 1071.2 | 690.4 | 745.3 | 23.1 | 24.0 | 26.6 | 27.2 | 21.8 | 22.3 |
| | 1963 | 2007 | 539.6 | 550.6 | 122.4 | 273.7 | 24.9 | 24.8 | 29.0 | 28.5 | 23.3 | 22.2 |
| | 1980 | 2012 | 1038.3 | 1029.0 | 662.8 | 591.2 | 23.2 | | 27.5 | | 22.2 | |
| | 1980 | 2002 | 1038.3 | 1012.8 | 662.8 | 668.6 | 23.2 | 24.4 | 27.5 | 28.8 | 22.2 | 23.2 |
| | 1964 | 2003 | 668.3 | 643.0 | 387.8 | 371.6 | 23.7 | 25.3 | 27.7 | 28.4 | 23.1 | 23.0 |
| Reference period vs. | 1966 | 2006 | 874.3 | 894.6 | 512.5 | 584.4 | 24.3 | 24.2 | 27.0 | 27.7 | 21.8 | 22.2 |
| fruit tree planting period | 1982 | 2005 | 978.3 | 984.3 | 669.0 | 651.0 | 23.3 | 23.7 | 26.8 | 29.2 | 21.6 | 23.9 |
| | 1976 | 2015 | 852.1 | 816.0 | 543.7 | 348.5 | 22.9 | | 26.3 | | 21.6 | |
| | 1969 | 2002 | 1025.1 | 1012.8 | 359.1 | 668.6 | 23.3 | 24.4 | 28.2 | 28.8 | 22.8 | 23.2 |
| | 1973 | 2005 | 991.1 | 984.3 | 760.3 | 651.0 | 23.5 | 23.7 | 26.9 | 29.2 | 22.4 | 23.9 |
| | 1969 | 2012 | 1025.1 | 1029.0 | 359.1 | 591.2 | 23.3 | | 28.2 | | 22.8 | |
| | 1985 | 2009 | 485.6 | 461.3 | 245.8 | 134.3 | 23.1 | 25.0 | 28.3 | 29.1 | 22.9 | 23.3 |

$P_w$ and $Q_w$ refer to precipitation and streamflow in the wet season, respectively.

**Table A4.** The pairs of rainfall events in the Jiujushui watershed.

| Period | Selected Rainfall Events | Paired Rainfall Events | Peak Rainfall | | Antecedent Rainfall (3-Day Average)/mm | | Time Interval/Day | |
|---|---|---|---|---|---|---|---|---|
| | **1976** 20 Apr–26 Apr | **1986** 24 Mar–29 Mar | 26.4 | 26.3 | 0 | 0.9 | 0 | 1 |
| | **1964** 16 May–22 May | **1987** 8 Sep–14 Sep | 33.9 | 33.4 | 0.8 | 1.3 | 0 | 0 |
| | **1964** 10 Jan–15 Jan | **1990** 19 Feb–25 Feb | 45 | 45.1 | 0.2 | 1.8 | 1 | 1 |
| | **1967** 18 May–23 May | **1991** 18 Mar–23 Mar | 52.5 | 51.6 | 0.1 | 0.4 | 0 | 1 |
| | **1962** 14 Apr–20 Apr | **1992** 27 May–2 Jun | 34.7 | 35 | 0.2 | 1.6 | 0 | 2 |
| | **1976** 10 Oct–16 Oct | **1993** 11 May–17 May | 32.6 | 32.9 | 0.2 | 0 | 1 | 0 |
| | **1976** 12 May–18 May | **1994** 7 May–13 May | 29.1 | 29 | 0.8 | 0 | 1 | 1 |
| Reference period vs. | **1961** 17 Nov–23 Nov | **1996** 16 Apr–22 Apr | 41 | 40.7 | 0.1 | 0 | 0 | 0 |
| reforestation period | **1972** 30 Oct–5 Nov | **1996** 19 Jul–25 Jul | 25.6 | 25.7 | 0.3 | 0.4 | 0 | 0 |
| | **1974** 28 Oct–3 Nov | **1996** 19 Jul–25 Jul | 25.6 | 25.7 | 0.3 | 0.4 | 0 | 0 |
| | **1968** 6 Jun–12 Jun | **1997** 28 Apr–4 May | 70.8 | 73.7 | 2.5 | 2.2 | 0 | 1 |
| | **1964** 16 May–22 May | **1998** 29 Oct–4 Nov | 33.9 | 33.5 | 0.8 | 0.1 | 0 | 1 |
| | **1965** 2 Aug–7 Aug | **1997** 11 Oct–17 Oct | 41.2 | 41.7 | 0 | 1.2 | 1 | 1 |
| | **1976** 27 Sep–3 Oct | **1987** 22 Sep–28 Sep | 39.6 | 38.7 | 0.5 | 0.2 | 0 | 1 |
| | **1984** 11 Nov–17 Nov | **1990** 30 Oct–5 Nov | 44.9 | 44.7 | 1.7 | 0 | 1 | 0 |
| Reference period vs. fruit tree planting period | **1983** 20 Aug–26 Aug | **2002** 11 May–16 May | 25.7 | 26.1 | 1.3 | 1 | 2 | 1 |
| | **1976** 20 Apr–26 Apr | **2014** 1 May–7 May | 26.4 | 26.5 | 0 | 0 | 0 | 1 |
| | **1964** 16 May–22 May | **2010** 11 Feb−17 Feb | 33.9 | 33.4 | 0.8 | 1.2 | 0 | 1 |
| | **1964** 11 Oct–17 Oct | **2016** 12 May−18 May | 44.4 | 44 | 0.2 | 0 | 0 | 0 |
| | **1966** 1 Apr–7 Apr | **2004** 9 Apr−14 Apr | 39.7 | 39.3 | 2.0 | 0 | 0 | 0 |
| | **1967** 18 May–23 May | **2016** 7 Sep−13 Sep | 52.5 | 52 | 0.1 | 0 | 0 | 1 |
| | **1968** 6 Jun–12 Jun | **2001** 28 Aug−3 Sep | 70.8 | 68.7 | 2.5 | 2.8 | 0 | 0 |
| Reference period vs. fruit | **1978** 25 Aug–31 Aug | **2005** 20 Apr−26 Apr | 30.1 | 30.1 | 0 | 0 | 1 | 0 |
| tree planting period | **1962** 14 Apr–20 Apr | **2006** 28 Apr−4 May | 34.7 | 34.6 | 0.2 | 0.7 | 0 | 1 |
| | **1976** 10 Oct–16 Oct | **2007** 1 Aug−7 Aug | 32.6 | 32.2 | 0.2 | 1.5 | 1 | 1 |
| | **1977** 24 Feb–1 Mar | **2006** 19 May−25 May | 40.2 | 40.7 | 0 | 0 | 0 | 0 |
| | **1975** 3 Jul– 10 Jul | **2005** 20 Apr−26 Apr | 30.3 | 30.1 | 0.2 | 0 | 1 | 0 |
| | **1972** 26 Apr–2 May | **2002** 27 May−2 Jun | 26.6 | 26.6 | 0.7 | 0 | 1 | 1 |
| | **1961** 17 Nov–23 Nov | **2006** 19 May−25 May | 41 | 40.7 | 0.1 | 0 | 0 | 0 |
| | **1965** 2 Aug–7 Aug | **2016** 14 Mar−19 Mar | 41.2 | 42 | 0 | 0 | 2 | 1 |

Time interval: time between rainfall peaking and flow peaking during a rainfall event.

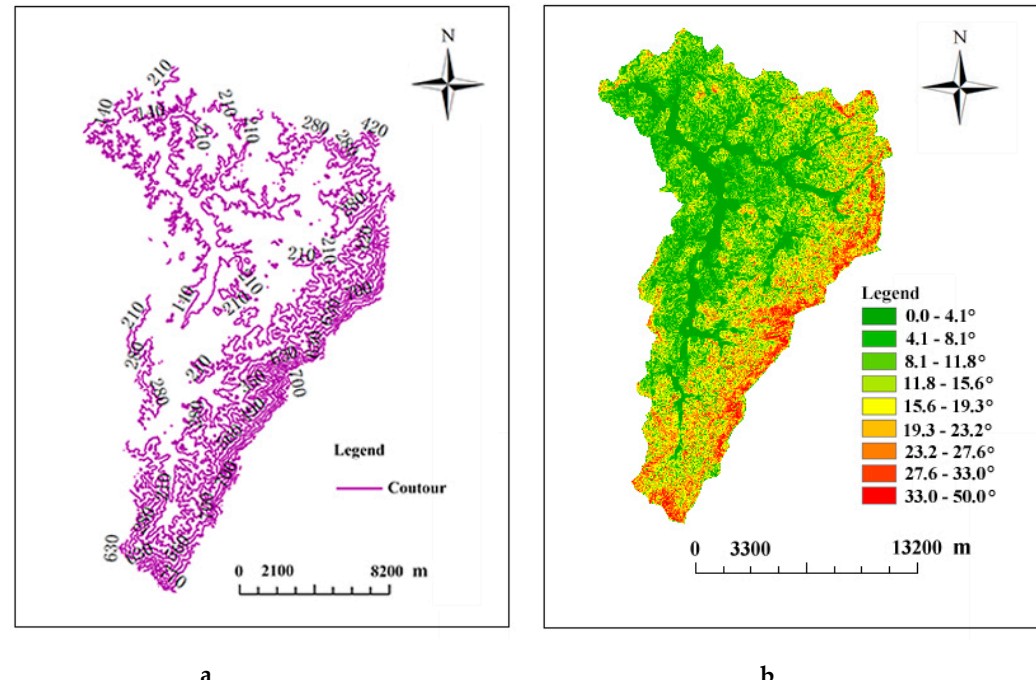

**Figure A1.** Watershed characteristics: (**a**) watershed elevation, and (**b**) watershed slope.

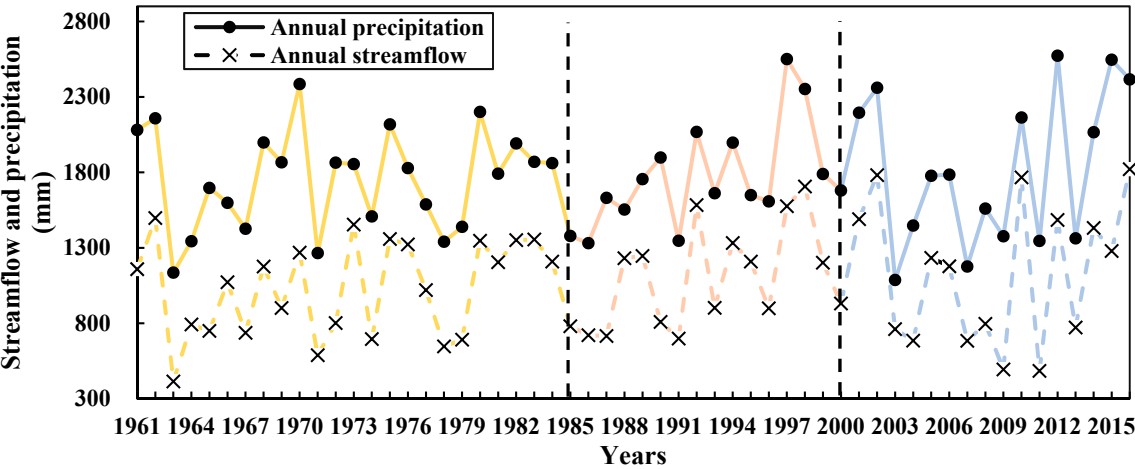

**Figure A2.** Annual precipitation and streamflow from 1961 to 2016 in the Jiujushui watershed (different colors represent three periods: reference, reforestation and fruit tree planting).

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
