# Peer review of "Contrasting Differences in Responses of Streamflow Regimes between Reforestation and Fruit Tree Planting in a Subtropical Watershed of China"

_forests, doi:10.3390/f10030212_

Round 1

Reviewer 1 Report

Thanks for the interesting manuscript. You've tried to consider most of the reviewers comments, that's why I can acknowledge an overall improvement. However - your conclusions are quite interesting when it comes to evaluating the possible (and according to your study non-existent) benefits of fruit tree plantings, but it still seems a little bit vague to me to adress just this monocausal conclusion (fruit trees as only explanation for the observed changes).

Author Response

Responses to the comments by reviewer 1

1. Thanks for the interesting manuscript. You've tried to consider most of the reviewer’s comments, that's why I can acknowledge an overall improvement. However - your conclusions are quite interesting when it comes to evaluating the possible (and according to your study non-existent) benefits of fruit tree plantings, but it still seems a little bit vague to me to address just this monocausal conclusion (fruit trees as only explanation for the observed changes).

Response: Thanks for the comment. The significant and negative impacts of fruit tree plantings on high flows are indeed interesting and surprising. Our common perception is that fruit tree plantings can produce benefits of both ecominic development and environmental protection. However, our study suggests fruit tree plantings can significantly increase high flows and consequently lead to greater soil erosion potentials. This unexpected result is reasonable as large-scale fruit tree plantings (about 25% of the whole watershed) can significantly disturb and compact soil due to intensive human activities. It should be noted that our study did not find significant changes on annual mean flow as well as low flow caused by fruit tree plantings.

In fact, our conclusion on the impacts of fruit tree plantings on high flows is conservative. During the period of fruit tree planting, our study watershed had experienced forest recovery from reforestation with other tree species, which should offset the hydrological effects of fruit tree plantings to some extent. Clearly, this offsetting effect was not strong enough so that the effects of fruit tree plantings on high flows were pronounced.

Reviewer 2 Report

I thank the authors for the effort in resubmitting the paper. The manuscript has been improved somewhat, but many of the shortcomings of the study remain. As long as these are clearly stated and described in the manuscript, I think the study has some value in being published.

Below, I have made note of the remaining issues as I see them, along with suggestions.

1.         OK

2.         There is still no real detail on the landuse across the catchment. There is no spatial map of landuse, showing the changes in land use during the 3 periods. There is still no mention of land use within the catchment besides forest and fruit trees. ie. Are their urban centres and other agricultural enterprises across the catchment?

3.         There is still no detail on soil type within the catchment, other than stating there is "red soil". Please include some information about the main soil types in the catchment and how they vary with depth.

4.         OK

5.         OK

6.         I still have reservations, given the magnitude and time taken for changes in forest cover and fruit trees, that any differences should be detectable.

7.         OK

8.         OK

9.         OK

10.     OK

11.     OK

12.     I am still unclear what Figure 6 is showing. Fig shows flows above a certain threshold, they do not show duration of flows as suggested in the text.

13.      OK

14.      OK

Author Response

Responses to the comments by reviewer 2

1. There is still no real detail on the landuse across the catchment. There is no spatial map of landuse, showing the changes in land use during the 3 periods. There is still no mention of land use within the catchment besides forest and fruit trees. ie. Are their urban centres and other agricultural enterprises across the catchment?

Response: Thanks for the further comments. Our study watershed is a forested watershed with forest cover being more than 70%. Based on the historical land use data, the major land cover types include forest land, farmland and town.  The changes in farmland and town only account for < 3.5% (1962-2006) and 0.2% (1996-2005) of the watershed area, respectively (Line 81-84). Thus, forest change is a dominating driver for streamflow variation.

Regarding a spatial map showing the changes in land use duing the 3 periods, we understand that this map should be useful for this paper. But unfortunately, such a map is not available for this study.

2. There is still no detail on soil type within the catchment, other than stating there is "red soil". Please include some information about the main soil types in the catchment and how they vary with depth.

Response: Thank the reviewer for pointing this out. To address this concern, we have added some data on soil types and their variations with soil depth (Line 73-77).

3. I still have reservations, given the magnitude and time taken for changes in forest cover and fruit trees, that any differences should be detectable.

Response: We understand the reviewer’s concern. However, the hydrological effects of forest change (reforestation and fruit tree planting) on flow regime are cumulative over time and space. In our study watershed, both reforestation and fruit tree planting cumulatively reached quite significant levels (>25% of the whole watershed) so some significant hydrological effects are logically expected. Please also see our response to the comment by reviewer 1. 

4. I am still unclear what Figure 6 is showing. Fig shows flows above a certain threshold, they do not show duration of flows as suggested in the text.

Response: In the section of materials and methods, flow duration refers to the average number of days when daily flows exceed or are below a median value (threshold) in all selected years (Line 139-142). As an example, Figure 6 shows that 18 ‘points’ in 1968 are below the threshold (1.93 m3s-1), so the flow duration in 1968 is 18 days, while 0 day in 1994 because all ‘points’ exceed the threshold. To reduce confusion, we have modified the text as well as the figure caption.

This manuscript is a resubmission of an earlier submission. The following is a list of the peer review reports and author responses from that submission.

Round 1

Reviewer 1 Report

Dear authors,
thanks for your manuscript - it was really interesting, that fruit trees, which were planted predominantly, have a negative effect on runoff dynamics even though it is "forested" land.

However, I have just a small question:

81 ff. I am just curious: Is there any information about forest coverage before 1961? As there is an increase of 40.7% between 1986 and 2016 but only a decrease of 6.3% (1961-1986) my question would be if genuinely new forested areas developed or if it was reforestation of former (before 1961) forest area.

Reviewer 2 Report

Review of “Contrasted differences in responses of streamflow regimes between reforestation and fruit tree planting in a sub-tropic watershed of China"

Journal: Forests

Manuscript ID: forests-404059

Authors: Zhipeng Xu, Wenfei Liu, Xiaohua Wei, Houbao Fan, Yizao Ge, Guanpeng Chen and Jin Xu

General comments

This manuscript describes a study to investigate effects of reforestation and fruit tree planting on flow regime in a catchment in southern China. The topic of this study is certainly of relevance to this journal, and the manuscript itself appears to be of suitable length, but the study lacks substantial shortcomings that would need to be addressed before it could provide meaningful insights into the effects of reforestation and fruit tree planting on flow regime.

I have provided some main comments and suggestions below, and have included comments in an attached PDF version of the manuscript. I feel this manuscript requires very major revision before being suitable for publication in Forests.

1.    For a paper to be published in Forests, I would expect there to be a certain level of detail surrounding the forests involved. This manuscript contains no detail of the species studied, where and how they were planted, or how they are managed.

2.    There is no detail on the landuse across the catchment. A spatial map of landuse, showing the changes in land use during the 3 periods would be useful. What are the other main land use besides forest and fruit trees? What type of forest was there before it was harvested during 1961-1985?

3.    There is no detail of the spatial characteristics of the catchment. How does rainfall vary across the catchment? What is the catchment's elevation, slope, aspect, soil type, depth, hydrogeology? This will all affect the nature of changes in streamflow regime.

4.    In several areas, value language is used eg. … "we conclude that reforestation had positive impacts on high and low flows …" It is not clear why the effects are positive effects. Please be clear why effects in this area are considered to be positive or negative.

5.    Similarly, the term 'forest changes' used, presumably to refer to changes in forest cover. Please be more specific, and make it clear if this refers to reafforestation or fruit trees.

6.    The period of data is broken into 3 to facilitate analysis, but the reality is that the changes in land use occurred during these periods. Below is an estimate from Fig 3. The reference period seems sensible, are there is relatively little change in forest and fruit tree cover. However, the reforeststion and fruit trees periods show a gradual

Period

Name

Change   % forest cover

Average   % forest cover

Change   % fruit trees

Average   %  fruit trees

1961-85

Reference/

Deforestation

43à37

~40

0à2

~1

1985-2000

Reforestation

37à70

~55

2à6

~4

2000-2016

Fruit   Trees

70à78

~74

6à28

~15

Table A2 shows the paired years. In several cases, years early in the reforestation period were used eg 1987-1990) during which time, forest cover had only changed by a few %. There is no way these can be compared (it is often said that it is difficult to detect changes in streamflow where the change in land use is less than 20% of the whole catchment). The same applies to fruit trees, where the first few years of fruit tree planting are used to make comparisons. Furthermore, many changes in hydrological regime only become evident after several years after land use has changed significantly.

7.    One of my biggest concerns is that comparisons made between the fruit tree vs deforestation period, includes not only a change in %cover of fruit trees (~15 compared with ~1%), but a much larger change in forest cover (~74 compared with ~40%).

8.    It does not make complete sense to take 2 similar years in terms of annual rainfall, and then compare individual events between those years. Events are likely to be affected by antecedent conditions, that extend back longer than 3 days. Eg. Information on soils may show that they can hold 100mm.

9.    It is difficult to infer changes in some of the metrics of streamflow regime, when only daily data is used. Many of these changes occur at finer resolution ie. hours.  For example, your finding of 0 days delay, may have been a delay of a few hours – if you had hourly data.

10.I am not clear how certain metrics of streamflow regime can be linked to land use. For example, "timing for low flow refers to the date of minimum flow occurrence in a year". This would be much more affected by rainfall during the year, and not much by land use.

11.It is not clear how rainfall events were selected. How many different rainfall events were examined? How were they selected? Was there a criterion they had to meet? Why did you choose 25-75 mm events only?

12.It is unclear what Figure 6 is showing. Fig shows flows above a certain threshold, they do not show duration of flows as suggested in the text. It is unclear what the points represent - daily flows?

13.In the results section, the text refers to differences in flow regime for the different land uses, but then states the differences are not significant. In that case, the differences do not exist, and the text should not suggest they do.

14.There are vague comments made in the discussion eg. increasing recharge can lead to low streamflows. That's why it is important to include information on catchment characteristics, so you can place your results into context and explain them better. Also, how does fruit tree establishment lead to reduced surface roughness and soil water holding capacity? You should provide details on this.

Reviewer 3 Report

The submitted article is an investigation into the role of reforestation of fruit trees on streamlow characteristics in China. I do feel that the topic, while not novel, is of great interest to the readers of Forests.

However, I must reject this manuscript based upon serious methodological flaws that invalidate the study. I present some of these below and have provided a document with more comments on the introduction and methods. I found the flaws in the methodology significant enough to warrant rejection and did not offer critique of the results and discussion for this reason.

The authors provided no information on the land use history of the watershed, other than a simple explanation of deforestation and reforestation. There must be some quantification of the forest cover, basal area, or LAI in order to interpret the results. Furthermore, the readers must be given information on HOW the data were collected. We don't even have any information on the species that are present on the landscape.

Second, there are significant flaws with the hydrologic analysis. It is extremely questionable to use climate data to interpret changes in hydrograph characteristics, especially when land use changes as extreme as deforestation and reforestation have been changing the runoff ratios in the watershed. Furthermore, we are told that the deforestation period is used as a reference. Given that deforestation is often followed by rapid recovery in streamflow, this is inappropriate. The climate variables used in the analysis are also poor indicators of streamflow response - a correlation of 78% is fairly low at high flows (we would expect >90%) and a temperature relationship with streamflow of 28% is simply too low to make any hydrolgoical inferences (even if it is statistically significant). Lastly, the authors incorrectly define exceedance probability, which makes me further question the analysis.

It appears to me that this manuscript is the result of an "opportunity" to look at streamflow in a gauged watershed rather than a rigorous analysis of reforestation on streamfow. It does not hold up scientifically and has no place in peer-reviewed literature that could be used to inform public policy.

The introduction, as currently written, is too short and contains very little synthesis of the literature regarding the impacts of re-forestation and deforestation on hydrologic characteristics. There are also minor grammatical errors throughout the text. Consider reading Sun et al. (2005), Forests and water relationships: hydrologic implications of forestation campaigns in China as a great reference to the available literature in the US, China, and abroad.

Methods – Why is the slope presented as 0.94 per mil?

Line 79 – missing a , after precipitation

Line 84 – How can you attribute the change in streamflow to deforestation? This must be done using a paired-watershed study or at least a more thorough analysis. Or you should cite a paper where this analysis was conducted.

Line 90 – You cannot use a deforestation period as a reference period, as streamflow is changing over time at this phase. Or more information about the actual deforestation and vegetation control strategies in order to attempt to make this claim.

Line 103 – You have incorrectly labeled Q5 and Q95 and your high/low flows. High flow should be the flow exceeded at Q95 and low flow should be the flow less than Q5.

Line 111 – Duration is typically used when referring to precipitation, precisely because high flows are usually in response to a rain event and low flows occur after extreme drying. I’m okay using your definitions as long as you put in a disclaimer such that they’re measuring two very different hydrologic processes.

Line 114 – Variability is not the same as the rate of change, especially with regard to trying to compare high and low flows

Line 117 – In hydrologic literature, Return Period is denoted as Tr, not R.

Section 2.5 – I understand the reasoning to use a paired-year approach for the analysis, likely because this was not established as a paired-watershed study. However, I do not think it’s appropriate in this study because the deforestation event altered the runoff ratios of the watershed. Additionally, in Table 2, the climate variables simply do not have a strong enough correlation to be used for an analysis of change in hydrograph properties pre- and post-reforestation.

Figure 2 – It is rather obvious that the temperature line has been drawn in to be a dashed line. It’s sloppy and should be updated.

Figure 3 – You haven’t provided any information on how these data were calculated. Without any knowledge of the sampling strategy, these are simply points on a line.

Table 1 – Length is ambiguous and should be changed to drainage length (or whatever it’s referring to)